# Mechanisms of EGFR-TKI-Induced Apoptosis and Strategies Targeting Apoptosis in EGFR-Mutated Non-Small Cell Lung Cancer

**DOI:** 10.3390/genes13122183

**Published:** 2022-11-22

**Authors:** Shigetoshi Nishihara, Toshimitsu Yamaoka, Fumihiro Ishikawa, Kensuke Higuchi, Yuki Hasebe, Ryo Manabe, Yasunari Kishino, Sojiro Kusumoto, Koichi Ando, Yusuke Kuroda, Tohru Ohmori, Hironori Sagara, Hitoshi Yoshida, Junji Tsurutani

**Affiliations:** 1Division of Gastroenterology, Department of Medicine, Showa University School of Medicine, Tokyo 142-8666, Japan; 2Advanced Cancer Translational Research Institute, Showa University, Tokyo 142-8555, Japan; 3Division of Respirology and Allergology, Department of Medicine, Showa University School of Medicine, Tokyo 142-8666, Japan; 4Center for Biotechnology, Showa University, Tokyo 142-8555, Japan; 5Tokyo Metropolitan Ebara Hospital, Tokyo 145-0065, Japan

**Keywords:** EGFR-TKI, apoptosis, EGFR-mutated NSCLC, combination therapies

## Abstract

Homeostasis is achieved by balancing cell survival and death. In cancer cells, especially those carrying driver mutations, the processes and signals that promote apoptosis are inhibited, facilitating the survival and proliferation of these dysregulated cells. Apoptosis induction is an important mechanism underlying the therapeutic efficacy of epidermal growth factor receptor (EGFR)-tyrosine kinase inhibitors (TKIs) for EGFR-mutated non-small cell lung cancer (NSCLC). However, the mechanisms by which EGFR-TKIs induce apoptosis have not been fully elucidated. A deeper understanding of the apoptotic pathways induced by EGFR-TKIs is essential for the developing novel strategies to overcome resistance to EGFR-TKIs or to enhance the initial efficacy through therapeutic synergistic combinations. Recently, therapeutic strategies targeting apoptosis have been developed for cancer. Here, we review the state of knowledge on EGFR-TKI-induced apoptotic pathways and discuss the therapeutic strategies for enhancing EGFR-TKI efficiency. We highlight the great progress achieved with third-generation EGFR-TKIs. In particular, combination therapies of EGFR-TKIs with anti-vascular endothelial growth factor/receptor inhibitors or chemotherapy have emerged as promising therapeutic strategies for patients with EGFR-mutated NSCLC. Nevertheless, further breakthroughs are needed to yield an appropriate standard care for patients with EGFR-mutated NSCLC, which requires gaining a deeper understanding of cancer cell dynamics in response to EGFR-TKIs.

## 1. Introduction

The discovery of somatic activating mutations in the epidermal growth factor receptor (*EGFR*) associated with dramatic responses to tyrosine kinase inhibitor (TKI) treatment resulted in precision medicine in the field of lung cancer therapeutics. Exon 19 deletions and exon 21 L858R point mutations in *EGFR* comprise the majority of EGFR-activating mutations, which are conserved and highly sensitive to EGFR-TKIs. These genetic alterations confer the ligand-independent constitutive activation of EGFR and lead to the activation of the major downstream signaling pathways, including PI3K/AKT and RAS/ERK1/2, thereby promoting cell survival and proliferation [1].

Treatment with the latest third-generation EGFR-TKI osimertinib, which is currently widely used in clinical settings, has been shown to improve median progression-free survival (PFS) to a greater extent than first-generation EGFR-TKIs (median PFS of 18.9 months vs. 10.2 months) [2]. Moreover, overall survival (OS) improved significantly with osimertinib treatment compared with that obtained using first-generation EGFR-TKIs (38.6 months vs. 31.8 months) [3]. There have been significant clinical advances for the treatment of patients with EGFR-mutated advanced non-small cell lung cancer (NSCLC). However, disease progression with EGFR-TKI treatment remains inevitable. Thus, the mechanisms of resistance to EGFR-TKI therapy must be understood to develop future therapeutics for patients with EGFR-mutated advanced NSCLC.

A major challenge to precision medicine is providing successful treatment for patients with advanced cancer. It is necessary to continue effective treatments sequentially by focusing on accurate molecular targets and/or to provide more effective therapeutics in first-line treatment to obtain a better response and longer OS in patients with NSCLC. Toward this end, it is important to find strategies for effectively enhancing mechanisms of cancer cell death, including apoptosis, with EGFR-TKI treatment.

Mechanistic insights into the induction of apoptosis through BCL-2 family members have contributed to the improved understanding of the biology underlying EGFR-TKI resistance. BCL-2 interacting mediator of cell death (BIM) activation is required for TKIs to induce apoptosis. Conversely, a decrease in BIM activity level is sufficient to confer resistance to EGFR-TKIs in EGFR-mutated NSCLC cells [4,5,6]. Furthermore, genetic loss of *BIM*, such as deletion polymorphisms, can limit the clinical efficacy of EGFR-TKI treatment [7]. Therefore, BIM might play a central role in EGFR-TKI-induced apoptosis. However, recent studies have revealed that EGFR-TKI-induced apoptosis is mechanistically more complicated. For example, along with the intrinsic apoptosis pathway (i.e., involving the BCL-2 family, physically associated with the mitochondria), the extrinsic apoptosis pathway, which involves death receptors, is suggested to participate in the induction of apoptosis by EGFR-TKIs [8,9]. Furthermore, promising combination therapies with potential molecular targeting agents that enhance apoptosis induction may be identified. Here, we discuss recent mechanistic insights into the induction of apoptosis by EGFR-TKIs in preclinical and clinical settings. Furthermore, we discuss current and future therapeutic strategies for enhancing the efficacy of EGFR-TKIs based on these mechanisms.

## 2. Effectiveness and Limitations of EGFR-TKI Therapy in EGFR-Mutated NSCLC

### 2.1. Efficacy and Limitations of Single EGFR-TKI Therapy

First-generation EGFR-TKIs (gefitinib and erlotinib), which reversibly bind to EGFR, and second-generation EGFR-TKIs (afatinib and dacomitinib), which irreversibly bind to EGFR, have shown marked antitumor activity in patients with EGFR-sensitizing mutations, with an objective response rate (ORR) of 61–83% [10,11,12]. Several randomized phase III trials comparing gefitinib or erlotinib to platinum-based chemotherapies for treatment-naïve patients with NSCLC and EGFR mutations demonstrated that these drugs significantly improve the PFS compared to that obtained with platinum-based chemotherapies [10,11,12,13]. However, resistance to EGFR-TKIs is inevitable and generally occurs within 12 months [10,11,12,13,14,15]. The most common EGFR-dependent resistance mechanism is the T790M secondary mutation in EGFR, which reduces the binding affinity of EGFR-TKIs to the TK domain. *MET* and *HER2* amplifications are representative mechanisms of EGFR-independent resistance, which act as bypass tracks [16,17]. Osimertinib, a third-generation EGFR-TKI that irreversibly binds to EGFR, was designed to overcome resistance due to the T790M mutation, and it selectively inhibits EGFR sensitizing and T790M resistance mutations, while avoiding wild-type EGFR inhibition [18]. A high ORR and promising PFS were observed in patients with NSCLC carrying the T790M mutation treated with osimertinib for whom first- or second-generation EGFR-TKIs failed [19,20]. Additionally, a randomized phase III trial assessing the superiority of osimertinib to first-generation EGFR-TKIs (gefitinib or erlotinib) was conducted in treatment-naïve patients with EGFR-mutated NSCLC [2]. Although the ORR was similar to that obtained with first-generation EGFR-TKIs (80% in the osimertinib group vs. 76% in the first-generation EGFR-TKI group), osimertinib achieved significantly longer PFS (18.9 months in the osimertinib group vs. 10.2 months in the first-generation EGFR-TKI group) and had a less severe toxicity profile [2]. Therefore, osimertinib is now recommended as the standard first-line single EGFR-TKI treatment for patients with EGFR-mutated NSCLC. However, drug resistance also occurs with osimertinib. EGFR-dependent mechanisms due to additional EGFR mutations, such as C797S, L792X, and L718X, have been reported to be involved in osimertinib resistance, along with multiple EGFR-independent mechanisms, including *MET* amplification, *HER2* amplification, gene reconstitution, PIK3CA mutation, mitogen-activated protein kinase (MAPK) signaling activation, small cell carcinoma transformation, and epithelial-mesenchymal transition [17]. Consequently, the efficacy of single EGFR-TKIs is limited owing to emerging resistance, highlighting the need for combination therapies or new therapeutic approaches.

### 2.2. Combined EGFR-TKI Therapy and Development of Novel Therapeutic Approaches

Preclinical studies have shown that some cytotoxic agents and EGFR-TKIs are synergistically active against EGFR-mutated lung cancer cells. Erlotinib treatment in combination with cisplatin showed the potential to inhibit cell growth in an EGFR-mutated lung cancer model [21]. A preclinical investigation indicated that pemetrexed treatment overcomes acquired resistance to gefitinib in EGFR-mutated lung cancer cells [22]. Carboplatin–pemetrexed administered concurrently with gefitinib greatly impacted the tumor response rate and exhibited a survival benefit over gefitinib alone in a randomized phase III trial [23]. Vascular endothelial growth factor (VEGF) plays a critical role in tumor growth and is closely associated with resistance to antineoplastic agents, including EGFR-TKIs [24]. Thus, anti-VEGF treatment has the potential to overcome EGFR-TKI resistance. Preclinical studies have suggested that EGFR-TKIs combined with the anti-VEGF antibody bevacizumab could enhance tumor activity in EGFR-TKI-resistant lung cancer cells [25,26]. A clinical trial conducted in Japan to evaluate the efficacy of the first-generation EGFR-TKI erlotinib combined with bevacizumab showed the superiority of this treatment over erlotinib alone with regard to PFS [27]. Another global clinical trial showed that erlotinib combined with the anti-VEGF-receptor 2 (VEGFR2) antibody ramucirumab significantly improved PFS compared with that obtained with erlotinib alone [28]. Therefore, EGFR-TKIs combined with platinum-based chemotherapy or anti-VEGF treatment are alternative therapies for overcoming single EGFR-TKI resistance. Moreover, the occurrence of T790M and C797C mutations in *trans* but not *cis* alleles was reportedly suppressed by the combination of first- and third-generation EGFR-TKIs in a preclinical study [29]. Some case reports also revealed that the combination of osimertinib and gefitinib (or erlotinib) resulted in clinical improvement after sequential single EGFR-TKI therapy for erlotinib (or afatinib)- and osimertinib-relapsed cases. Although these cases are rare, the clonal progression of T790M/C797S from *in trans* to *in cis* was reported as a potential mechanism of resistance to combination therapy with first- and third-generation EGFR-TKIs [30,31]. In summary, clinical trials for novel combination therapies with EGFR-TKI are required to further confirm the efficacy observed in preclinical studies.

## 3. Signaling Pathways of Apoptosis

Apoptosis is a type of cell death characterized by unique morphological changes (membrane blebbing, chromatin condensation, nuclei fragmentation, and cell shrinkage), and its mediators are dysregulated in many types of cancers [32]. Apoptosis is mediated by the caspase family, which is composed of 12 cysteine proteinases in humans, six of which are proteolytically activated in response to apoptotic signals [33]. Apoptotic caspases can be classified as initiator (caspases 8, 9, and 10) and effector (caspases 3, 6, and 7) caspases. Apoptosis occurs through two main mechanisms—intrinsic and extrinsic pathways—both of which lead to the activation of initiator caspases and subsequent effector caspase activation (Figure 1). Ultimately, effector caspases induce the activation of endonuclease by degrading the inhibitor of caspase-activated DNase (ICAD) and other proteases, which results in the degradation of chromosomal DNA and nuclear and cytoskeletal proteins, leading to cytomorphological changes, such as chromatin and cytoplasmic condensation as well as nuclear fragmentation [34]. Caspases are directly or indirectly suppressed by inhibitor of apoptosis (IAP) family proteins (e.g., X-linked IAP [XIAP], cIAP1, and cIAP2), which are overexpressed in many human tumors and are thus promising targets for cancer therapy [35].

### 3.1. Intrinsic Apoptosis Pathway

The intrinsic pathway is triggered by diverse intracellular stresses, such as DNA damage, nutritional stress, and the removal of growth factors, including chemotherapeutic agents or molecular target inhibitors, and is characterized by mitochondrial outer membrane permeabilization (MOMP) [36]. MOMP results from the formation of pores in the mitochondrial outer membrane (MOM), which allows the release of cytochrome c from the mitochondrial intermembrane space (IMS) into the cytosol. Released cytochrome c binds to and activates apoptotic peptidase activating factor-1 (APAF-1) in the cytosol, forming an oligomerized protease complex called the apoptosome in the presence of ATP. The apoptosome cleaves and yields mature caspase 9 to elicit the subsequent caspase cascade, which initiates apoptosis. Along with cytochrome c, other types of apoptotic proteins, such as second mitochondria-derived activator of caspase (SMAC) and high-temperature requirement protein A2 (HTRA2) are also released from the IMS following MOMP, which promotes caspase activation by neutralizing IAP family proteins, thus inducing apoptosis [37].

MOMP is an irreversible step in the apoptosis process and is strictly regulated by a balance between the activity of anti-apoptotic proteins and pro-apoptotic BCL-2 family proteins (Table 1) [38]. The central effectors of the pro-apoptotic BCL-2 family proteins BAK and BAX are activated by other pro-apoptotic BCL-2 family proteins classified as BCL-2 homology 3 domain (BH3)-only proteins, which are subdivided into activators (e.g., BIM, BID, and PUMA) and sensitizers (e.g., BAD, NOXA, BIK, BMF, and HRK). Activators directly bind to and activate BAK/BAX, resulting in conformational changes, oligomerization, and consequent pore formation in the MOM, resulting in MOMP [39]. In contrast, sensitizers indirectly activate BAK/BAX by neutralizing the effects of anti-apoptotic proteins (e.g., BCL-2, BCL-XL, and MCL1), which bind to and inhibit the activation of BAK/BAX [40].

BCL-2 family proteins are transcriptionally and post-translationally regulated in a specific manner by specific molecules. For example, BAX, BID, PUMA, and NOXA expression is upregulated by the p53 tumor suppressor protein to induce apoptosis [41,42,43]. In response to survival factors, BAD is phosphorylated at two serine residues by survival kinases, including AKT, which enables the binding of BAD to 14-3-3 proteins in the cytosol to block the association with BCL-2 and BCL-XL [44,45]. Similarly, BIM is also phosphorylated by ERK1/2 MAPK, which mediates its dissociation from BCL-XL and MCL1, and possibly degradation by the ubiquitin-proteasome system [46,47]. These unique activation mechanisms for individual pro-apoptotic proteins can block the activity of various stress signals that typically trigger the intrinsic apoptosis pathway.

### 3.2. Extrinsic Apoptosis Pathway

The extrinsic apoptosis pathway is triggered by activation of transmembrane proteins known as death receptors, including the FasL receptor Fas, tumor-necrosis factor (TNF) receptors TNFR1 and TNFR2, and TNF-related apoptosis-inducing ligand (TRAIL) receptors DR4 and DR5 [48,49,50]. Upon ligand binding, death receptors are trimerized through the intracellular death domain and form a death-inducing signaling complex (DISC) by recruiting the adaptor protein Fas-associated protein with death domain (FADD) and caspases 8 and 10. Following DISC formation, dimerized caspases 8 and 10 are activated by autocleavage, which in turn activates effector caspases, leading to apoptosis [51]. In addition to its key role in effector caspase activation, caspase-8 cleaves BID proteins, so that truncated BID (tBID) is myristoylated and translocated to the mitochondria to promote MOMP in a BAX-dependent manner [52]. Accordingly, BID acts as a bridge between the intrinsic and extrinsic pathways. The requirement of BID cleavage for the execution of apoptosis is the main characteristic discriminating between two cell types. In type 1 cells, such as thymocytes, activation of effector caspases by caspase-8 is sufficient for apoptosis induction by TRAIL and FasL, indicating the dispensability of the tBID-mediated intrinsic pathway [53]. In contrast, BID cleavage is required for the induction of apoptosis in type 2 cells, such as hepatocytes and pancreatic cells [54,55]. Currently, it is thought that the expression level of XIAP discriminates between type 1 and type 2 cells because genetic deletion of *XIAP* or treatment with SMAC mimetics can convert type 2 cells into type 1 cells [56]. In this model, effector caspases are directly inhibited by XIAP proteins in type 2 cells, even if they are activated by caspase-8. However, when SMAC proteins are released from the mitochondria following tBID-mediated MOMP, effector caspases are relieved from XIAP and contribute to apoptosis [57].

## 4. Mechanistic Insights into EGFR-TKI-Induced or -Enhanced Apoptosis from Preclinical Models

### 4.1. Pro-Apoptotic Protein: BIM

BIM is a BH3-only activator of the pro-apoptotic BCL-2 protein family [58] (Table 1). Currently, BIM is thought to play a central role in the induction of apoptosis via EGFR-TKIs. BIM has been recognized as an essential factor for hematopoietic cell homeostasis and cytokine deprivation-induced apoptosis [59]. Moreover, imatinib, targeting the TKI BCR–ABL, induces apoptosis via BIM and, to a lesser extent, BAD in BCR–ABL-positive leukemia cells [60]. Several preclinical studies in EGFR-mutated NSCLC cells suggested that BIM is essential for the apoptosis triggered by EGFR-TKI treatment [4,5,6,61]. These studies consistently reported that BIM expression is upregulated by treatment with an EGFR-TKI (e.g., gefitinib or erlotinib), with EGFR-TKI-induced apoptosis observed in EGFR-TKI-sensitive cell lines. Conversely, attenuation of BIM expression using small interfering RNA conferred resistance to EGFR-TKIs, even in EGFR-TKI-sensitive cells. Furthermore, BIM expression is regulated and influenced by the MEK-ERK1/2 signaling cascade acting downstream of EGFR, but not by the PI3K/AKT or JNK pathway [4,5]. Gong et al. [5] indicated that, in addition to upregulated expression of BIM by EGFR-TKI in apoptosis induction, the effector pro-apoptotic protein BAX displays altered localization from the nucleus to the cytoplasm (likely the mitochondria) in erlotinib-treated apoptotic H3255 lung cancer cells. The distribution of cytochrome c changed from punctate to diffuse, consistent with the release of cytochrome c from the mitochondria [5]. Furthermore, overexpression of the anti-apoptotic protein BCL-XL in PC-9 NSCLC cells blocked erlotinib-induced apoptosis, suggesting that EGFR-TKI-induced apoptosis is mediated predominantly by the intrinsic pathway [5].

The BH3-mimetic ABT-737 acts as a small-molecule BCL-2/BCL-XL antagonist with antitumor activity [62]. In the BCR–ABL translocated cell line K562, ABT-737 showed a relatively minimal effect as a single agent, whereas its combined treatment with imatinib significantly enhanced the cell-killing activity [60]. Similarly, ABT-737 alone had a minimal effect on EGFR-mutated cells, such as PC-9, H3255, and H1650 cells, whereas apoptosis induction was greatly enhanced in combination with EGFR-TKIs [4,5]. Therefore, the combination of EGFR-TKIs and BCL-2/BCL-XL antagonists is a potential therapeutic strategy for the treatment of EGFR-mutated NSCLC.

In general, the MEK-ERK1/2 pathway suppresses BIM expression by direct phosphorylation through the proteasomal degradation of BIM [63,64]. Suppression of the MEK-ERK1/2 pathway by EGFR-TKIs results in the upregulated expression of BIM. Therefore, BIM upregulation appears to be essential for the induction of apoptosis but is not sufficient for apoptosis in EGFR-mutated cancer cells. Interestingly, the integral role of BIM in promoting apoptosis in response to molecular targeted therapies has been determined in BRAF-mutant colorectal cancer cells [65], BRAF-mutated melanoma cells [66], BCR–ABL-positive leukemia cells [67], and EML4-ALK-positive NSCLC cells [68]. Furthermore, high pretreatment levels of *BIM* RNA strongly induced the apoptosis of EGFR-mutated, HER2-amplified, and PI3CA-mutated cancer cells when subjected to EGFR, HER2, and PI3K inhibitor treatment, respectively; however, there was no association of the *BIM* pretreatment level with the responsiveness to cytotoxic chemotherapeutic agents, such as Taxol, gemcitabine, and cisplatin [69]. Conversely, lower pretreatment *BIM* RNA levels were associated with the weak apoptotic activity of EGFR-TKIs in NSCLC cell lines. A lower *BIM* expression level was also significantly related to shorter PFS in patients with EGFR-mutated NSCLC [69]. Similarly, in the EURTAC study, patients treated with erlotinib who had low or intermediate levels of *BIM* mRNA expression had significantly worse median PFS than those with high levels of *BIM* (7.2 vs. 12.9 months; *p* = 0.0003) [70].

*BIM* deletion polymorphisms have been detected in approximately 12–15% of patients with NSCLC [71,72,73]; these mutations have been described in Asian and Hispanic populations, with no cases in African American or Caucasian populations. These deletions lead to alternative splicing, which results in BIM isoforms lacking the BH3-domain. Expression of these isoforms is associated with a poor clinical response to EGFR-TKIs [74] and a lower median PFS [7,75]. Accordingly, there have been extensive research efforts to overcome *BIM* deletion-related resistance to EGFR-TKIs. Xia et al. [71] reported that ABT-737 and BH3 mimetics enhanced BIM expression, and that the combination of erlotinib and ABT-737 remarkably enhanced the induction of apoptosis and suppressed the growth of xenograft tumors in EGFR-mutated NSCLC cells with *BIM* deletion (HCC2279 cells). Nakagawa et al. [76] determined that vorinostat, a histone deacetylase (HDAC) inhibitor, also increased the BIM expression level, and that the combination of gefitinib and vorinostat enhanced the induction of apoptosis in PC-3 cells with *BIM* deletion [76]. To our knowledge, no functional studies have been conducted on the interactions between *BIM* deletion and angiogenesis. Clinically, EGFR-TKIs plus bevacizumab confer a significantly higher ORR and median PFS in patients with advanced EGFR-mutated NSCLC and *BIM* deletion [77]. Furthermore, Liu et al. [78] demonstrated that EGFR-TKIs plus chemotherapy (pemetrexed, gemcitabine plus cisplatin, or carboplatin) significantly improve ORR and prolong median PFS and OS compared with EGFR-TKIs alone in patients with EGFR-mutated NSCLC and *BIM* deletions. Therefore, the combination of EGFR-TKIs and drugs to increase BIM expression, such as chemotherapy, ABT-737, or HDAC inhibitors, may overcome resistance to EGFR-TKI alone with *BIM* deletion polymorphisms or low BIM expression. Taken together, because BIM is essential for EGFR-TKI-induced apoptosis, *BIM* mRNA expression or *BIM* deletion polymorphisms critically affect the effectiveness of EGFR-TKI treatment.

### 4.2. Anti-Apoptotic Proteins: BCL-2, BCL-X, and MCL1

BCL-2 family members antagonize each other to regulate cell death and survival. Overexpression of anti-apoptotic proteins and/or low expression of pro-apoptotic proteins can enhance survival and drug resistance [79]. Increased expression of BCL-2 and BCL-XL has been observed in NSCLC cell lines and specimens from patients [80,81]; these proteins participate in drug resistance resulting from the suppression of apoptosis induction by mitochondrial dysfunction [82,83,84]. However, the mechanism whereby increased BCL-2 expression is associated with EGFR-TKI resistance in EGFR-mutated NSCLC has not yet been determined.

The loss-of-function genetic alteration in the splicing factor RNA-binding motif 10 (*RBM10*) limits the initial response to EGFR-TKIs in EGFR-mutated NSCLC by suppressing apoptosis induction [85]. RBM10 regulates the alternative splicing of apoptosis-related genes, such as *BCL-X* [86]. Using small hairpin RNAs (shRNAs), the attenuation of RBM10 expression led to diminished EGFR-TKI-induced apoptosis by decreasing the ratio of BCL-XS (pro-apoptotic isoform) to BCL-XL (anti-apoptotic isoform), resulting in increased BCL-XL expression relative to BCL-XS expression in the EGFR-mutated NSCLC cell lines PC-9 and H3255. Therefore, inhibition of BCL-XL combined with EGFR mutations can overcome the resistance to drugs, such as navitoclax and osimertinib, induced by RBM10 deficiency.

Myeloid cell leukemia 1 (MCL-1) is frequently overexpressed and amplified in human cancers, which is associated with a poor prognosis [87,88]. MCL-1 expression was found to be upregulated via EGFR signaling, and EGF stimulation transcriptionally enhanced *MCL-1* expression through the transcription factor Elk-1 [89]. In EGFR-mutated NSCLC cells, the increased expression levels of MCL-1 mRNA and protein were found to be mediated by mTORC1-hyperactivation to confer EGFR-TKI resistance [90]. Therefore, EGFR-TKIs can inhibit PI3K-mTOR signaling, which leads to the downregulation of MCL-1 expression, ultimately significantly inducing apoptosis in EGFR-mutated NSCLC cells [91]. Interestingly, the decreased levels of ERK1/2-dependent BIM and MCL-1 play a major role in osimertinib-induced apoptosis through proteasomal degradation. Moreover, in cells with acquired drug resistance, MCL-1 expression was not decreased by osimertinib, indicating that decreased expression of MCL-1 is essential for osimertinib-induced apoptosis [79]. The combination of osimertinib, inhibition of MCL-1, and activation of BAX enhanced apoptosis induction and inhibited the growth of osimertinib-acquired resistant cells both in vitro and in vivo [92]. Hence, targeting apoptosis using agents that target members of the BCL-2 family could be a promising therapeutic strategy for overcoming the acquired resistance to EGFR-TKIs.

### 4.3. Extrinsic Pathway in EGFR-TKI-Induced Apoptosis

In 2011, Bivona et al. [93] determined that 36 genes reproducibly confer erlotinib sensitivity in H1650 cells, which are less sensitive to erlotinib and harbor an *EGFR* exon 19 deletion and *PTEN* loss. To identify shRNAs that could restore the sensitivity of H1650 cells to erlotinib, a pooled shRNA library targeting >2000 cancer-relevant genes was introduced into H1650 cells. Among these 36 genes, 18 were associated with nuclear factor kappa-light-chain-enhancer of activated B cell (NF-κB) signaling, which is recognized as a survival signaling cascade. Silencing of the major NF-κB subunit RELA, intermediate signaling proteins linked to FAS/NF-κB, such as cellular FADD-like IL-1b-converting enzyme (FLICE)-inhibitory protein (c-FLIP) and RIPK, and FAS enhanced erlotinib sensitivity, regardless of *PTEN* loss. Furthermore, these effects were observed in other EGFR-mutated cells, such as 11–18, HCC827, and H3255 cells. Interestingly, the reduction in cell viability was associated with increased caspase 3/7 activity, suggesting that silencing these genes enhances erlotinib-induced apoptosis. Because the NF-κB pathway is inhibited through IκB kinase (IKKβ) inhibition, using the pharmacological inhibitor of IKKβ BMS-345541 or knockdown of IKKβ expression by shRNA suppressed NF-κB phosphorylation through an increase in IKβ expression, thereby restoring erlotinib sensitivity and inducing apoptosis in H1650 cells. Clinically, reduced IKβ expression levels (resulting in a high NF-κB activation state) were associated with worse PFS and decreased OS in patients with EGFR-TKI-treated EGFR-mutated NSCLC. Thus, FAS and NF-κB signaling appear to modulate the induction of apoptosis by EGFR-TKI treatment [93].

c-FLIP is a truncated version of caspase-8, the key regulator of caspase-8 activity, and acts as an inhibitor of the extrinsic apoptotic pathway [94]. Based on a study by Bivona et al. [93], Shi et al. [8] revealed that osimertinib and other EGFR-TKIs reduce the expression level of c-FLIP by facilitating proteasomal degradation in primarily sensitive EGFR-mutated cell lines, such as PC-9, HCC827, and H1975 cells. Conversely, overexpression of c-FLIP partially conferred resistance to osimertinib. c-FLIP suppression was thus suggested to regulate the antitumor activity of EGFR-TKIs in EGFR-mutated NSCLC [8].

TRAIL, a member of the TNF superfamily, can induce apoptosis in cancer cells when bound to its associated death receptors, DR4 and DR5 [95]. The DR4 expression level was decreased by treatment with EGFR-TKIs (e.g., osimertinib) in EGFR-mutated NSCLC cells and was associated with EGFR-TKI-induced apoptosis. Increased DR4 expression was also identified in acquired resistant EGFR-mutated cells or tissue samples of patients with recurrence following acquired resistance to EGFR-TKIs. Conversely, forced suppression of DR4 enhanced osimertinib-induced apoptosis and restored the drug sensitivity. DR4 attenuation via EGFR-TKIs was found to be mediated by DR4 proteasomal degradation through membrane-associated RING-CH-8 (MARCH-8) and by inhibition of MEK/ERK/AP-1-dependent *DR4* transcription. Clinically, DR4 expression is significantly associated with poorly differentiated tumors rather than with well- or moderately differentiated lung adenocarcinoma tumors. Moreover, patients with DR4-positive lung adenocarcinoma had significantly worse OS than those with DR4-negative tumors [96]. Therefore, a connection may exist between EGFR-TKI-induced apoptosis and DR4 modulation.

We reported that BID cleavage through caspase-8 activation is involved in the induction of apoptosis by EGFR-TKIs in PC-9 cells and cells resistant to osimertinib, in which resistance is mediated by bypassing from EGFR signaling to IGF1R activation [9]. In these cells, osimertinib treatment upregulated BIM expression; however, BID cleavage was observed only in PC-9 cells and not in the resistant cells. When the cells were treated with osimertinib and the IGF1R inhibitor linsitinib, BID was cleaved, which induced apoptosis. Thus, caspase-8 activation is required for the cleavage of BID in PC-9 and resistant cells. Cleaved BID was translocated to the mitochondria from the cytosol in PC-9 cells treated with osimertinib and in cells resistant to osimertinib and linsitinib. These findings suggested that BIM elevation alone is not sufficient to induce apoptosis in NSCLC cells, which further requires the cleavage of BID and its translocation to the mitochondria through caspase-8 activation [9]. These factors suggested the involvement of the extrinsic apoptotic pathway in EGFR-TKI-induced apoptosis in vitro and in vivo.

### 4.4. Other Mechanisms by which EGFR-TKIs Induce or Enhance Apoptosis

TNF regulates the anti-apoptotic and pro-apoptotic signaling pathways in a balanced manner. TNF induces stress-activated protein kinase/c-Jun NH2-terminal kinase, caspases 3 and 8, and p38 MAPK, which are all involved in pro-apoptotic pathways [97,98,99]. In contrast, the anti-apoptotic pathways induced by TNF include the PI3K/AKT, ERK/MAPK, and NF-κB pathways [97,98,99]. The activation of PI3K/AKT by TNF is mediated by crosstalk with EGFR. EGFR-TKIs inhibit the TNF-mediated activation of PI3K/AKT signaling through EGFR and its downstream IAPs, thereby blocking the anti-apoptotic pathway and activating the pro-apoptotic pathway (via TNF, caspases 3 and 8, and others) [97]. Moreover, TNF expression levels are increased in response to EGFR inhibitors, such as erlotinib, through NF-κB activation, which has been observed in EGFR-mutated and unmutated NSCLC cell lines, xenograft models, and archival tissue samples from patients. In EGFR-mutated HCC827 and H3255 cells, TNF upregulation conferred a partially protective effect against EGFR-TKI exposure. Therefore, blocking TNF with etanercept or thalidomide enhanced the effectiveness of EGFR-TKIs [100].

SMAC proteins, which are released from the mitochondria into the cytoplasm during apoptosis, antagonize the function of IAP proteins [101,102,103,104]. Focusing on this effect, SMAC mimetics have been developed to neutralize cIAP1, cIAP2, and XIAP to increase the sensitivity of cancer cells to apoptosis. These SMAC mimetics were found to enhance the sensitivity of NSCLC cells to multiple chemotherapeutic agents [101,103]. In addition, clinical phase I and II trials targeting IAPs using SMAC mimetics are underway [104,105,106].

p53 affects EGFR-TKI-induced apoptosis through activation of the Fas/FasL-mediated signaling pathway, indicating that p53 is required for enhanced EGFR-TKI sensitivity [107,108,109]. In contrast, p53 mutations attenuate EGFR-TKI sensitivity [107,108].

Table 2 lists the proteins associated with EGFR-TKI-induced apoptosis via the intrinsic and extrinsic apoptosis pathways.

Autophagy may confer protection from stress conditions, such as those induced by chemotherapy, including EGFR-TKIs. EGFR activation suppresses autophagy and promotes NSCLC growth via beclin-1 phosphorylation [110]. Conversely, the EGFR-TKI erlotinib induces autophagy possibly through EGFR/beclin-1 complex disruption and promotes the initiation of autophagy in EGFR-mutated NSCLC cells (HCC827) [110]. The induced autophagy may lead to chemoresistance [111]. In the EGFR-mutated NSCLC cells, HCC827 and HCC4006, erlotinib induces autophagy, and the autophagy inhibitor, chloroquine, enhances erlotinib sensitivity. Furthermore, silencing of *atg5* or beclin-1, which are autophagy-related genes, also significantly increases erlotinib sensitivity [112]. Therefore, autophagy inhibition may enhance EGFR-TKI activity. Beclin-1 contains a BH3 domain that binds to anti-apoptotic proteins, such as BCL-2, BCL-XL, or MCL-1 [113]. This interaction suppresses beclin-1 activity and reduces the initiation of autophagy. Phosphorylation of the BH3 domain leads to decreased interaction between beclin-1 and BCL-2 family proteins, resulting in the release of beclin-1 and its subsequent activation to initiate autophagy [114,115,116]. Although interaction between beclin-1 and BCL-2 family proteins is required to regulate apoptosis and/or autophagy, mechanistic insights regarding the modulation of EGFR-TKI-induced apoptosis in EGFR-mutated NSCLC cells are lacking. Another significant approach to reduce EGFR-TKI-induced apoptosis involves hypoxia-inducible factor (HIF)-1a. HIF-1a levels are higher in EGFR-TKI-acquired resistance tissue samples than those before receiving EGFR-TKI treatment [117]. Chronic gefitinib treatment promotes reactive oxygen species (ROS) production and mitochondrial dysfunction and results in ROS-mediated EGFR-TKI resistance through epithelial mesenchymal transition [118]. Similarly, chronic oxidative stress has been connected to afatinib resistance [119]. Using genomic analysis, the expression of EMT markers, such as VIM and ZEB1, was correlated with HIF1a in lung adenocarcinoma samples. Moreover, there was a correlation between HIF1a and FGFR1 mRNA expression under hypoxic conditions [120]. FGFR1 expression was induced by hypoxia in the EGFR-mutated cells, H1975 and HCC827, which conferred osimertinib resistance through attenuation of BIM expression [120]. Therefore, hypoxia is a key microenvironmental stress associated with EGFR-TKI resistance and apoptosis. Another unique possible mechanism to induce an anti-apoptotic effect involves initiating apoptotic stress to promote the survival of neighboring cells. Under apoptosis stress, cells release fibroblast growth factor (FGF) 2, which results in the upregulation of the pro-survival proteins BCL-2 and MCL1 through MEK-ERK signaling [121]. High *FGF2* mRNA expression and FGFR1 amplification were observed in osimertinib-acquired resistance tumor tissue samples [122]. This may suggest a non-cell autonomous EGFR-TKI resistance through FGF2. Taken together, EGFR-TKI-induced apoptosis is regulated by BCL-2 family proteins, and modulation of homeostatic circumstances might be required to conquer EGFR-mutated NSCLC.

## 5. Clinical Translation Potential of the Interaction of EGFR-TKIs with Apoptosis

Recently, Su et al. [123] reported a meta-analysis of the impact of *BIM* polymorphisms on the treatment effect of EGFR-TKIs among 14 trials, including 2649 patients with EGFR-mutated NSCLC who had been treated with EGFR-TKIs, such as gefitinib, erlotinib, and afatinib. The ORR and disease control rate (DCR) of the 487 patients (18.08%) with *BIM* polymorphisms were inferior (odds ratio [OR] = 0.49, 95% confidence interval [CI]: 0.34–0.70, *p* < 0.001; OR = 0.50, 95% CI: 0.30–0.84, *p* = 0.009, respectively), and these patients also had a shorter OS (in the Korean and Taiwan subgroups, hazard ratio [HR] = 1.34, 95% CI: 1.18–1.53, *p* < 0.001; in the other country subgroups, HR = 2.43, 95% CI: 2.03–2.91, *p* < 0.001) than patients without *BIM* polymorphisms did [123]. Similar results were reported by Lv et al. [124] in a meta-analysis of 3003 patients with EGFR mutation treated with gefitinib, erlotinib, afatinib, and icotinib.

The EURTAC trial is a phase III study that showed a significant benefit of erlotinib compared to standard chemotherapy in patients with EGFR-mutated NSCLC [13]. This trial showed that patients with high *BIM* mRNA expression had longer PFS upon erlotinib treatment (12.9 months; 95% CI: 9.7–23.9) than patients with low/intermediate *BIM* mRNA expression did (7.2 months; 95% CI: 2.6–12.3). Moreover, in a univariate analysis, survival in the high *BIM* mRNA expression group was 28.6 months (95% CI: 19.8–not reached), while survival in the low/intermediate *BIM* mRNA expression group was 22.1 months (95% CI: 14.7–30.3, *p* = 0.0364). Thus, high *BIM* expression was associated with longer survival (HR = 0.53, 95% CI: 0.30–0.95, *p* = 0.0323). Conversely, low *BIM* expression was associated with a poor clinical response to erlotinib among patients with EGFR-mutated NSCLC (Table 3).

Several clinical studies have been conducted to overcome the EGFR-TKI resistance mediated by enhanced BIM activity and expression. A phase I trial was performed for the HDAC inhibitor vorinostat in combination with gefitinib in 12 patients with EGFR-mutated NSCLC and *BIM* polymorphisms [125]. Dose-limiting toxicity (DLT) was not observed in dose escalation of vorinostat, and 400 mg/day vorinostat every other week plus 250 mg/day gefitinib daily was recommended for the phase II trial. The 12 patients had a DCR of 83.3% and median PFS of 5.2 months (Table 4). However, firm conclusions of the clinical benefit of this combination could not be made given the small number of patients enrolled in this trial [125]. Interestingly, EGFR-TKI plus bevacizumab treatment in patients with EGFR-mutated advanced NSCLC and *BIM* deletion polymorphisms resulted in a significantly higher ORR (94.4% vs. 40%; *p* > 0.001), longer PFS (11.12 months vs. 7.87 months; *p* = 0.001), and prolonged OS (30.9 months vs. 25.4 months; *p* = 0.06) compared to those of patients receiving EGFR-TKI treatment alone (Table 5) [77]. Moreover, in patients with EGFR-mutated NSCLC and *BIM* deletion polymorphisms, the combination of EGFR-TKI and pemetrexed or gemcitabine-based platinum doublet chemotherapy significantly improved the ORR (65.5% vs. 38.9%; *p* = 0.046) and prolonged the PFS (7.2 months vs. 4.7 months; *p* = 0.008) and OS (18.5 months vs. 14.2 months; *p* = 0.107) compared to EGFR-TKI treatment alone (Table 5) [78]. In these studies, combination therapies with bevacizumab or chemotherapy conferred significantly higher ORR, longer PFS, and longer OS. Although these are retrospective analyses with small sample sizes, these treatments comprise a promising approach for patients with EGFR-mutated NSCLC and *BIM* deletion polymorphisms.

In hematologic malignancies, BH3-mimetics (e.g., navitoclax [ABT-263], obatoclax [GX15-070], venetoclax [ABT-199]) have been assessed in clinical trials to increase the effective induction of malignant cell apoptosis and decrease the emergence of drug resistance [128]. Recently, a phase IB study of osimertinib and navitoclax in advanced EGFR-mutated NSCLC was reported [127]. A trial of 40–80 mg/day osimertinib plus 150–325 mg/day navitoclax was conducted in 27 patients with EGFR-mutated NSCLC (including those with T790M-positive tumors) previously treated with EGFR-TKIs. No DLTs were observed during the dose escalation. The ORR was 100%, and the PFS was 16.8 months (Table 5). Therefore, the combination of osimertinib and navitoclax seems to be clinically effective. Combination with BCL-1/BCL-XL inhibitors may be a promising strategy to enhance osimertinib activity; however, this warrants further study and validation.

## 6. Conclusions

We have provided a comprehensive review of the state of knowledge on the mechanisms by which EGFR-TKIs induce apoptosis, along with summarizing current information on clinical trials for patients with EGFR-mutated NSCLC. *BIM* deletion polymorphisms and low *BIM* mRNA expression confer intrinsic resistance to EGFR-TKI in patients with EGFR-mutated NSCLC [13,123]. This highlights the importance of gaining a deeper understanding of the mechanisms of EGFR-TKI-induced apoptosis and EGFR-TKI resistance to enable the development of more effective and targeted therapies. Furthermore, based on this review, we have proposed potential strategies for enhancing the therapeutic efficacy of EGFR-TKIs. In particular, for patients with EGFR-mutated NSCLC, novel therapeutic strategies to overcome drug resistance are required. The third-generation EGFR-TKI osimertinib has shown superiority with respect to PFS and OS compared to first-generation EGFR-TKIs [2]. Moreover, combination therapies of EGFR-TKIs with anti-VEGF(R) inhibitors or chemotherapy show promise as therapeutics for patients with EGFR-mutated NSCLC to maximize survival [27,28,129]. Over the past 20 years, gefitinib, a first-generation EGFR-TKI, has been used for the treatment of patients with NSCLC in Japan [130]. Many preclinical and clinical studies have accumulated extensive knowledge on these inhibitors. However, effective treatment in patients with EGFR-mutated NSCLC remains a challenge, as effectively targeting the molecules and pathways conferring resistance requires elucidating the diverse and complex mechanisms of EGFR-TKI resistance. Thus, breakthroughs in EGFR-mutated NSCLC therapies are sorely needed. To this end, the precise mechanisms underlying EGFR-TKI-induced apoptosis should be elucidated, and potent therapies regulating sensitivity to EGFR-TKIs must be developed. Further endeavors are necessary to yield an appropriate standard care for patients with EGFR-mutated NSCLC. In this regard, gaining a deeper fundamental understanding of cancer cell dynamics in response to EGFR-TKIs would be of great help.

## Figures and Tables

**Figure 1 genes-13-02183-f001:**
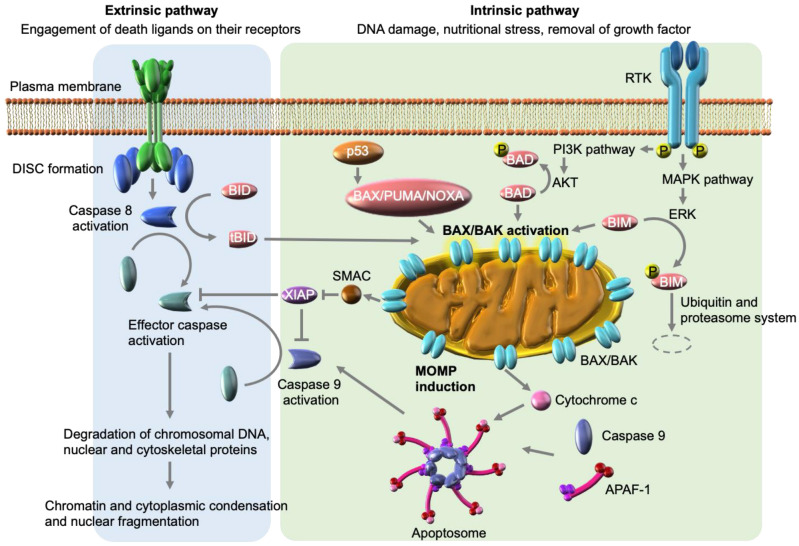
Intrinsic and extrinsic signaling pathways of apoptosis.

**Table 1 genes-13-02183-t001:** Functional classification of BCL-2 family proteins.

Classification		Mechanisms	BCL-2 Family Proteins
Anti-apoptotic proteins		Sequester of pro-apoptotic proteins	BCL-2, BCL-XL, MCL-1
Pro-apoptotic proteins			
	effectors	Induction of MOMP	BAX, BAK
	activators	Direct activation of BAX/BAK Inhibition of anti-apoptotic proteins	BIM, BID, PUMA
	sensitizers	Inhibition of anti-apoptotic proteins	BAD, NOXA, BIK, BMF, HRK

MOMP: mitochondrial outer membrane permeabilization.

**Table 2 genes-13-02183-t002:** Proteins in the intrinsic and extrinsic apoptosis pathways associated with EGFR-TKI-induced apoptosis.

Classification	Mechanism	Proteins Associated with EGFR-TKI-Induced Apoptosis
Intrinsic apoptosis pathway	Pro-apoptotic	BIM, BAX, BID, BCL-XS
	Anti-apoptotic	BCL-XL, MCL-1
Extrinsic apoptosis pathway	Pro-apoptotic	FAS
	Anti-apoptotic	c-FLIP, IKKb, DR4, TNF, IAPs

**Table 3 genes-13-02183-t003:** Association of *BIM* mRNA expression with the outcome of patients with EGFR-mutated NSCLC treated with erlotinib in the EURTAC study.

BIM mRNA Expression	Treatment	Patients (*n* = 42)	ORR	PFS: Month (95% CI)	*p*	OS: Month (95% CI)	*p*
high	Erlotinib	16	87.5%	12.9 (9.7–23.9)	0.0122 (HR = 0.49)	24.5 (14-NR)	0.323 (HR = 0.53)
low/intermediate	Erlotinib	26	34.6%	7.2 (2.6–12.3)		20.8 (8.9–32.5)	

**Table 4 genes-13-02183-t004:** Results of early phase IB clinical trials targeting apoptosis with EGFR-TKIs.

	Treatment	Patients	DCR (%) (95% CI)	PFS: Month (95% CI)
Takeuchi et al. [126]	Gefitinib (250 mg) + Vorinostat (400 mg)	12 ^1^	83.3% (0.52–0.98)	5.2 (1.4–15.7)
Bertino et al. [127]	Osimertinib (80 mg) + Navitoclax (150 mg)	27 ^2^	100%	16.8 (3.5-NR)

^1^ EGFR-activating mutation + BIM deletion polymorphisms, ^2^ EGFR-activating mutations, including EGFR T790M mutation.

**Table 5 genes-13-02183-t005:** Promising combination therapies with EGFR-TKIs for NSCLC patients with EGFR-activating mutations and BIM deletion polymorphisms.

	Treatment	Patients	ORR (%) (95% CI)	*p*	PFS: Month (95% CI)	*p*	OS: Month (95% CI)	*p*
Liu et al. [78]	EGFR-TKIs ^1^	36	38.9	0.046	4.7	0.008	14.2	0.107
	EGFR-TKIs ^1^ + platinum-based chemotherapy ^2^	29	65.5		7.2		18.5	
Cardona et al. [77]	EGFR-TKIs	15	40 (15.2–64.7)	<0.001	7.87 (7.5–10.5)	0.001	25.4	0.06
	EGFR-TKIs + Bevacizumab	18	94.4 (83.8–100)		11.12 (9.83–14.5)		30.9	

^1^ Gefitinib or erlotinib, ^2^ pemetrexed or gemcitabine plus cisplatin or carboplatin.

## Data Availability

Not applicable.

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
