# Peer review of "Mechanisms of EGFR-TKI-Induced Apoptosis and Strategies Targeting Apoptosis in EGFR-Mutated Non-Small Cell Lung Cancer"

_genes, 2022, doi:10.3390/genes13122183_

Round 1

Reviewer 1 Report

General Critique of Work

Minor issues to be addressed:

-Sentence line 343 is not clear, H1650 cells are sensitive then resistant!

-Authors would precise that TKI of first generation are reversible in comparison to the last one.

-Likewise, authors have to precise that depending on the allelic context of EGFR mutation, sensitivity to TKI remain impact independently of the expression of pro-apoptotic proteins (PMID: 25964297)

Major issues to be addressed:

-The role of mitochondria in apoptosis remains crucial; however, in context of hypoxia cancer cells trigger mitophagy limiting ROS generation and cell death. Because apoptosis and autophagy remain together at the crosstalk between cell survival or cell death (apoptotic or autophagic), that would interesting to address a short issue on this point and the role of BH3 only protein in this balance.

-Likewise, it was previously shown that EGFR is able itself to deactivate autophagy through Beclin phosphorylation and that TKI treatment impair this mechanism reactivating thereby apoptosis. It will interesting to discuss about this point. 

Author Response

Responses to the comments of Reviewer #1:

Minor issues to be addressed:
1)    Sentence line 343 is not clear, H1650 cells are sensitive then resistant!
Response:
We thank you for the valuable comment. Although H1650 cells harbor the EGFR-activating mutation exon 19 deletion, H1650 cells are less sensitive to EGFR-TKI because they harbor PTEN loss, which confers AKT activation (Chin TM et al. Clin Cancer Res.,2008: PMID: 18980981). We have revised the relevant sentence, which has been highlighted in yellow (Line 352).

2)    Authors would precise that TKI of first generation are reversible in comparison to the last one.
Response:
We agree with your comment. We have revised the relevant sentence, which has been highlighted in yellow (Lines 84, 85, and 96).

3)    Likewise, authors have to precise that depending on the allelic context of EGFR mutation, sensitivity to TKI remain impact independently of the expression of pro-apoptotic proteins.
Response:
We agree with your comment. The combination therapy with first and third generation EGFR-TKIs can inhibit trans T790M and C797S mutations in EGFR. Therefore, this should be included in section 2.2. We have revised the relevant sentence, which has been highlighted in yellow (Lines 139–147).

Major issues to be addressed:
4)    The role of mitochondria in apoptosis remains crucial; however, in context of hypoxia cancer cells trigger mitophagy limiting ROS generation and cell death. Because apoptosis and autophagy remain together at the crosstalk between cell survival or cell death (apoptotic or autophagic), that would interesting to address a short issue on this point and the role of BH3 only protein in this balance.
Response:
We agree with your comment. It would be interesting to discuss the crosstalk between apoptosis and autophagy, while modulating the induction of apoptosis in EGFR-mutated NSCLC cells by EGFR-TKI exposure. To the best of our best knowledge, our understanding of how apoptosis is regulated through the modulation of autophagy in EGFR-mutated NSCLC cells is poor. Almost all the reviewed studies employed wild-type EGFR-expressing cell lines because our publication is focused on the enhancement of apoptosis by EGFR-TKI in EGFR-mutated NSCLC. We have added a paragraph on autophagy and the hypoxic condition in EGFR-mutated NSCLC cells, which confers EGFR-TKI resistance, and discussed the involvement of BCL-2 family proteins in lines 437-475, which have been highlighted in yellow.

5)    Likewise, it was previously shown that EGFR is able itself to deactivate autophagy through Beclin phosphorylation and that TKI treatment impair this mechanism reactivating thereby apoptosis. It will interesting to discuss about this point.
Response:
We agree with your comment. It would be interesting to discuss the crosstalk between apoptosis and autophagy, while modulating the induction of apoptosis in EGFR-mutated NSCLC cells by EGFR-TKI exposure. To the best of our best knowledge, our understanding of how apoptosis is regulated through the modulation of autophagy in EGFR-mutated NSCLC cells is low. Almost all the reviewed studies employed wild-type EGFR-expressing cell lines because our publication is focused on the enhancement of apoptosis by EGFR-TKI in EGFR-mutated NSCLC. We have added a paragraph about autophagy and the hypoxic condition in EGFR-mutated NSCLC cells, which confers EGFR-TKI resistance, and discussed the involvement of BCL-2 family proteins in lines 437-475, which have been highlighted in yellow.

Thank you again for considering publication of our revised report in Genes. We believe that addressing the Editors and Reviewers’ comments has substantially improved the quality and impact of the manuscript. Please do not hesitate to contact me if you have further questions or points of clarification.

Reviewer 2 Report

In the present manuscript the authors provide an extensive and informative review of the literature focused on the Mechanisms of  the EGFR-TKI-induced apoptosis in the case of the EGFR mutated NSCLC. 

The paper is well designed and logically written. In the beginning, the authors explain the limitations of the single EGFR-TKI therapy and the superiority and efficacy of combined EGFR-TKI treatment with chemotherapeutic or anti-VEGF therapies. In the next section, they describe in detail the intrinsic and extrinsic apoptotic pathway before the main sections 4.1-4.4 in which they present the main proteins involved in the previously mentioned apoptotic pathways and their role in the EGFR-TKI resistance/sensitivity. 

There are a few aspects in which the manuscript could be improved:

-A table summarizing the results of the literature in the sections 4.1-4.4.

-A second table of the main clinical studies of Section 5.

Author Response

Responses to the comments of Reviewer #2:

“In the present manuscript the authors provide an extensive and informative review of the literature focused on the Mechanisms of the EGFR-TKI-induced apoptosis in the case of the EGFR mutated NSCLC. 
The paper is well designed and logically written. In the beginning, the authors explain the limitations of the single EGFR-TKI therapy and the superiority and efficacy of combined EGFR-TKI treatment with chemotherapeutic or anti-VEGF therapies. In the next section, they describe in detail the intrinsic and extrinsic apoptotic pathway before the main sections 4.1-4.4 in which they present the main proteins involved in the previously mentioned apoptotic pathways and their role in the EGFR-TKI resistance/sensitivity.”
Response:
We thank you for your positive evaluation of our manuscript.

Comments:
1)    A table summarizing the results of the literature in the sections 4.1-4.4.
Response:
We thank you for pointing this out, and we agree with this suggestion. Therefore, Table 2 has been added to the manuscript.

2)    A second table of the main clinical studies of Section 5.
Response:
       Thank you for the valuable comment. We have added Tables 3–5 to summarize the clinical studies of section 5.

Thank you again for considering publication of our revised report in Genes. We believe that addressing the Editors and Reviewers’ comments has substantially improved the quality and impact of the manuscript. Please do not hesitate to contact me if you have further questions or points of clarification.

Reviewer 3 Report

In this review article, Nishihara and colleagues discuss the pro-apoptotic effects of EGFR-TKI in NSCLC. They dissect the complexity and causes and resistance mechanisms of different generation of TKI at the molecular level. Moreover, they mention clinical trials that aim at improving TKI pro-apoptotic effect by combinatorial strategies.

The topic is certain interesting and timely. The great promise of the targeted therapy has been partially halted by the occurrence of genetic resistance. However, with a rational approach and novel studies, it is possible to delay the occurrence of resistance. This review captures well the new advances in the understanding of success and failure of targeted therapies. The authors focused on NSCLC, but the concepts discussed in this review article can be extended to other cancer types as well. Thereby, the article can be of interest to a larger audience.

My opinion is that the review is already good for publication, but it can further benefit from additional improvements.

1)      Some of the subsections just present a list of citations, without a conclusion/closure at the end of each subsection. Adding a short conclusion at the end of each subsection would improve the reading.

2)      Some recent publications have shown the non-autonomous (e.g. paracrine) effect of apoptosis that sends pro-survival signals to the neighbors, and interestingly this happens though EGFR-MAPK pathway (10.1016/j.devcel.2021.05.007) or FGF (10.1038/s41467-021-26613-0). This effect might cause resistance or poor response to therapies. This idea was first proposed several decades ago (10.1038/1781391a0), but it hasn’t received much attention till very recently. I think that the authors should discuss the possibility that resistance or poor response to therapy can emerge from the communication between apoptotic and the neighboring cells.      

Author Response

Responses to the comments of Reviewer #3:

“In this review article, Nishihara and colleagues discuss the pro-apoptotic effects of EGFR-TKI in NSCLC. They dissect the complexity and causes and resistance mechanisms of different generation of TKI at the molecular level. Moreover, they mention clinical trials that aim at improving TKI pro-apoptotic effect by combinatorial strategies.
The topic is certain interesting and timely. The great promise of the targeted therapy has been partially halted by the occurrence of genetic resistance. However, with a rational approach and novel studies, it is possible to delay the occurrence of resistance. This review captures well the new advances in the understanding of success and failure of targeted therapies. The authors focused on NSCLC, but the concepts discussed in this review article can be extended to other cancer types as well. Thereby, the article can be of interest to a larger audience.”
Response:
We thank you for your positive evaluation of our manuscript.

1)    Some of the subsections just present a list of citations, without a conclusion/closure at the end of each subsection. Adding a short conclusion at the end of each subsection would improve the reading.
Response:
Thank you for pointing this out and we agree with this. We have added related sentences in lines 146–147, 309–311, 405–406, and 472–475, which have been highlighted in yellow.

2)    Some recent publications have shown the non-autonomous (e.g. paracrine) effect of apoptosis that sends pro-survival signals to the neighbors, and interestingly this happens though EGFR-MAPK pathway (10.1016/j.devcel.2021.05.007) or FGF (10.1038/s41467-021-26613-0). This effect might cause resistance or poor response to therapies. This idea was first proposed several decades ago (10.1038/1781391a0), but it hasn’t received much attention till very recently. I think that the authors should discuss the possibility that resistance or poor response to therapy can emerge from the communication between apoptotic and the neighboring cells.
Response:
We thank you for the valuable comment. The emergence of communication between apoptotic cells and neighboring cells is a highly interesting issue worthy of attention because FGF2-FGFR1 signaling may be a potential bypass signaling pathway from EGFR-TKI.   We have added related sentences in lines 466–472, which have been highlighted in yellow.

Thank you again for considering publication of our revised report in Genes. We believe that addressing the Editors and Reviewers’ comments has substantially improved the quality and impact of the manuscript. Please do not hesitate to contact me if you have further questions or points of clarification.
